Long-term alfalfa planting mediates the coupling of soil water and organic carbon storage in a semi-arid area of the Loess Plateau, China

Ma Yuanyuan 1 2
Zhou Xiaoping 2 3
Shen Yan myynczx@163.com 1 4
Ma Hongbin 1 4
Fu Bingzhe 1 4
Lan Jian 1 4
1 College of Forestry and Prataculture, Ningxia University , Yinchuan , China
2 Ningxia Rural Science and Technology Development Center , Yinchuan , China
3 School of Community for Chinese Nation, North Minzu University , Yinchuan , China
4 Northern  Yanchi  Desert  Steppe  Observation and Research  Station  of  Ningxia , Yanchi , China
Brygadyrenko Viktor
Electronic publication date: 2024 Nov 5
Publication date: 2024
Volume: 12
Electronic Location ID: e18373
Received 2024 Aug 20; Accepted 2024 Sep 30
Copyright: ©2024 Ma et al.
Copyright year: 2024
Copyright holder: Ma et al.
License: This is an open access article distributed under the terms of the Creative Commons Attribution License, which permits unrestricted use, distribution, reproduction and adaptation in any medium and for any purpose provided that it is properly attributed. For attribution, the original author(s), title, publication source (PeerJ) and either DOI or URL of the article must be cited.
License URL: https://creativecommons.org/licenses/by/4.0/

Keywords: Alfalfa, Revegetation, Soil water storage, Soil organic carbon storage, Coupling, Long-term planting

Funding: Key R&D Program Project of Ningxia No. 2023BCF01022 Natural Science Foundation of the Ningxia Autonomous Region, China 2024AAC03413 This work was supported by the Key R&D Program Project of Ningxia (No. 2023BCF01022), and Natural Science Foundation of the Ningxia Autonomous Region, China, grant number 2024AAC03413. The funders had no role in study design, data collection and analysis, decision to publish, or preparation of the manuscript.

==============================
The key to restoring arid and semi-arid ecosystems is maintaining soil water and organic carbon contents. Alfalfa (Medicago sativa L.) is a high-yield perennial forage crop and performs ecological functions as a drought-resistance leguminous herb. It has been widely planted for reconstruction of degraded soils in the Loess Plateau in northwestern China, but long-term planting may affect soil carbon–water coupling and lead to crop yield reduction. To maximize the benefits of reconstructed grassland, this study explored the couplings of soil water, organic carbon, and alfalfa productivity along a reconstruction chronosequence in a semi-arid area of the Loess Plateau. Space-for-time substitution approach was used to select different-aged stands of reconstructed grassland (1, 5, 7, 10, 15, 20, 30 years old). Alfalfa above-ground biomass (AGB), soil water storage (SWS), organic carbon storage (SOCS), and carbon–water coupling coordination degree (D) were measured in the 0–100 cm soil profile. Alfalfa AGB reached a peak in the 7th year, and the degradation began in the 10th year. Both SWS and SOCS varied nonlinearly with stand age. The greatest loss of SWS occurred in the 15th year (80–100 cm depth), whereas the largest increase of SOCS occurred in the 30th year (0–20 cm depth). There was a negative feedback relationship between AGB and SWS over the 30-year study period (Pearson r = −0.835, P = 0.098). AGB and SOCS initially showed a trade-off within the first 10 years (Pearson r = −0.7431, P = 0.2569), in contrast to their positive feedback in the 20–30th years (Pearson r = 0.9978, P = 0.0421). A decoupling between SWS and SOCS (D < 0.6) was observed after 12 years of alfalfa planting. For agricultural production, a greater supply of water and organic fertilizer is required from the 7th year of alfalfa planting, and reseeding may be needed around the 10th year to prolong the life of alfalfa community. Alfalfa should be planted for no more than 12 consecutive years in the study area for ecological protection.

Introduction

The Loess Plateau located in the arid and semi-arid climatic zone of China suffers from the most serious soil erosion in the world (Liu et al., 2017). The key to restoring degraded soil in this region is maintaining soil, water, and organic carbon contents (Conant, Paustian & Elliott, 2001). Reconstructed grasslands can effectively promote ecosystem restoration and enhance soil carbon sequestration capacity (Peng et al., 2019). However, irrational reconstructed grassland leads to the drying of soil moisture, which could diminish vegetation productivity and hinder carbon sequestration in the long term (Zhang et al., 2018; Li et al., 2021).

Alfalfa (Medicago sativa L.) is a high-yield and high-quality perennial forage crop with outstanding economic benefits (Brink, Sanderson & Casler, 2015). It is also a drought-resistance leguminous herb that performs ecological functions such as soil conservation and carbon sequestration (Sim, Brown & Teixeira, 2017; Wang et al., 2017). Given its multiple advantages, alfalfa has been widely planted for forage production and ecological restoration in the Loess Plateau region (Huang et al., 2018). However, alfalfa has an extensive taproot system, which can take up water from deep soil layers and consume much more soil water than other forage grasses (Sim, Brown & Teixeira, 2017; Ren & Huang, 2016). This leads to conspicuous contradictions between ecological restoration and agricultural production, with the age of reconstructed alfalfa grasslands being an essential factor for the balance (Huang et al., 2018; Wang et al., 2015).

Soil water storage (SWS) is temporally stable in arid and semi-arid areas under various land-use types (Wang et al., 2015). During alfalfa planting, dynamic variation occurs in SWS, which is the key factor limiting ecosystem productivity in the Loess Plateau region (Huang & Shao, 2019). On the one hand, high water consumption by alfalfa, which has the extensive taproot system that can take up water from deep soil layers and consume much more soil water than other forage grasses (Sim, Brown & Teixeira, 2017; Ren & Huang, 2016), may have a negative impact on ecosystem service provision; on the other hand, alfalfa growth is likely to be suppressed by water deficits and aggravate soil desiccation over time (Wei et al., 2022). Owing to the great water demand of the plant, continuous planting of alfalfa causes declines in shallow groundwater levels (Zheng et al., 2012), and it takes a long time to recover soil water conditions (Ge et al., 2022). Additionally, the age of reconstructed alfalfa grasslands has a profound influence on the above-ground productivity (Huang et al., 2018). Therefore, it is essential to ascertain the optimal grassland age for maintaining a high alfalfa forage yield and preventing over-consumption of soil water by alfalfa.

Soil organic carbon (SOC) storage (SOCS) is the largest terrestrial carbon pool, which more than doubles atmospheric carbon storage. A minimal change in SOCS could affect atmospheric CO2 concentrations (McSherry & Ritchie, 2013). Grassland degradation in Loess Plateau has a noticeable negative impact on global carbon cycle and climate change considering its vast area (Li et al., 2008). Maintaining SOCS during the course of grassland reconstruction has aroused interest in scientific research nowadays (Han et al., 2011; Wang et al., 2015). Reconstruction with alfalfa can help to improve the carbon storage capacity of soil, which depend on grassland age (Cai et al., 2021). This calls for research to explore the dynamics of SOCS in the reconstructed alfalfa grasslands along an age gradient.

Since SWS and SOCS interact with each other, their coupling affects ecological processes in the soil (Liu, Dong & Ren, 2010; O’Brien et al., 2009; Kerr & Ochsner, 2020). SWS is regarded as the most important factor driving soil carbon cycling in grassland ecosystems (Wiesmeier, Urbanski & Hobley, 2018). In arid and semi-arid areas, even a slight increase in the SWS of topsoil increases the SOC (Wang, Wu & Zhu, 2014), whereas a higher SOCS level in turn contributes to the SWS (Wang et al., 2015). Both SWS and SOCS play a significant role in alfalfa productivity (Wood et al., 2016), which contributes a lot to the local economy (Wei et al., 2022). In order to improve the stability and sustainability of alfalfa grassland ecosystems, it is pertinent to unravel the patterns and driving mechanisms of soil carbon–water coupling under vegetation restoration. So far, the ecological impact of reconstruction with alfalfa on soil carbon–water coupling in arid and semi-arid areas has been rarely described (Wang et al., 2015; Li et al., 2021).

In the present study, we characterized the temporal dynamics of SWS and SOCS distribution along a 30-year chronosequence of alfalfa planting. The aims of the present study were to: (1) unravel the vertical distribution (0–100 cm) of SWS and SOCS in relation to alfalfa productivity across different age of reconstructed alfalfa grassland, (2) clarify the patterns of soil carbon–water coupling under reconstruction with alfalfa, and (3) ascertain the optimal age of alfalfa grassland for maintaining a high alfalfa forage yield and great ecological balance. Results of this study can be useful for the sustainable management of reconstructed alfalfa grasslands in arid and semi-arid ecosystems in the long term.

Materials & Methods

Study area and experimental setup

The study area is located in Longde County (35°21′–35°47′N, 105°48′–106°15′E), Guyuan City, southern Ningxia, China. As part of the Loess Plateau, this area has an elevation between 1,900–2,500 m above sea level. It is situated in a typical temperate continental climatic zone, with a mean temperature of 7.6 °C. The mean annual sunshine duration and frost-free period are 2200 h and 140–160 days, respectively. This area receives a mean annual precipitation of ∼433.6 mm (mainly in June–September) and its mean annual potential evaporation is 1360.6 mm. The major soil type is gray cinnamon soil.

Thirty years ago, there was almost no vegetation in the study area. Since the policy of returning farmland to forest or grassland was implemented in 1992, ecological restoration has begun in this area, and planting alfalfa is one of the most common restoration practices (Liu et al., 2020). The alfalfa cultivar used in the study area was Gannong No. 4. Based on space-for-time substitution, seven stands were selected along an age gradient (i.e., 1, 5, 7, 10, 15, 20, 30 years old) of reconstructed grassland with consistent elevation, slope, and aspect. All stands were managed under the same practices and spaced ∼500 m apart from each other. A stand of natural grassland (control, mainly Stipa bungeana) was selected at a distance of ∼800 m away from the reconstructed grassland. In each stand, five 1 m × 1 m quadrats were selected at random for field survey and sampling.

Determination of alfalfa productivity

The vegetation communities in each quadrat were surveyed from April to May 2022, which is the first harvest time of local alfalfa. All alfalfa plants per quadrat were harvested to measure plant height, fresh weight, dry weight, and stem-to-leaf ratio. The above-ground parts of the alfalfa plants were oven-dried at 75 °C for 72 h and then weighed to determine the above-ground biomass (AGB). The growth rate of AGB was calculated by the following equation (Huang et al., 2018): (1) R=A/N

where R is the growth rate of AGB (gm−2yr−1), A is the increase of AGB (gm−2) in a given age group compared with the younger age group, and N is the number of years of alfalfa planting.

Soil sampling and analysis

In each quadrat, three soil profiles were excavated using a 4-cm diameter and 20-cm-long soil auger. Soil samples were collected from depths of 0–100 cm at 20 cm intervals, and a total of 600 samples were obtained. The samples were weighed immediately after removing large stones and leaf litter by hand. Then, the samples were brought back to the laboratory, where they were oven-dried at 105 °C for 24 h and weighed again to determine gravimetric soil water content. To measure soil bulk density, intact samples were collected using 5-cm diameter cutting rings and oven-dried at 150 °C until constant weight. The potassium dichromate oxidation–external heating method was used for SOC analysis (Nelson & Sommers, 1982). SWS and SOCS were calculated as follows (Deng, Sweeney & Shangguan, 2013; Huang et al., 2018):

(2) SWSmm=SWC×BD×H×10−1

(3) SOCS=SOC×BD×H×10−2.

In Eq. (2), SWS is soil water storage (mm), SWC is gravimetric soil water content (%), BD is soil bulk density (g cm−3), and H is soil depth (cm). In Eq. (3), SOCS is soil organic carbon storage (kg m−2) and SOC is soil organic carbon content (g kg−1).

Evaluation of soil carbon–water coupling

The coupling degree model is a quantitative tool that measures the coordination between different systems and elements (Peng, Song & Zeng, 2011). The soil carbon–water coupling degree model was built as follows (Liang et al., 2022; Cong, 2019): (4) C=4fx⋅gxfx+gy2k

where C is the degree of soil carbon–water coupling (0 ≤ C ≤ 1). When C is close to 1, it indicates a well-coupled relationship between SWS and SOCS; when C is 0, it means a decoupled state of SWS and SOCS. K is the adjustment coefficient (k = 2 in this study) (Li et al., 2021). F(x) and g(y) are efficacy functions for comprehensive evaluation of SOCS (Eq. (5)) and SWS (Eq. (6)), respectively.

(5) fx= ∑i=1paxi

(6) gx= ∑j=1qbyj

where i and j are the soil layers for SOCS and SWS, respectively; a and b are the weights of SOCS and SWS in corresponding soil layers, respectively; xi and xj are the normalized values of SOCS and SWS in corresponding soil layers, respectively. The maximum–minimum method was adopted for data normalization (Huang et al., 2018).

By adding the comprehensive regulation index to the coupling degree model, the coupling coordination degree model was built as follows (Peng, Song & Zeng, 2011):

(7) D=C⋅T

(8) T=αfx+βgy

where D is the coordination degree of soil carbon–water coupling. A D value lower than 0.6 means an unbalanced coupling, whereas a higher D value indicates a more coordinated coupling (Liang et al., 2022). T is the comprehensive regulation index, which characterizes the overall coordination effect of SWS and SOCS. α and β are the weight coefficients of SWS and SOCS, respectively (α = β = 0.5 in this study, which means soil water maintenance and SOC accumulation are equally important).

Statistical analysis

All measurement data are expressed as mean ± standard deviation (n = 5). SPSS v27.0 (SPSS Inc., Chicago, IL, USA) was used to conduct two-way analysis of variance (ANOVA) followed by the least significant difference (LSD) tests for multiple comparisons. Figures were drawn using Sigmaplot v15.0 (Systat Software Inc., San Jose, CA, USA) and PyCharm v4.0.5 (JetBrains., Prague, Czech Republic).

Results

Alfalfa productivity and vegetation succession

After planting, alfalfa productivity in terms of plant height, stem diameter, dry weight, fresh weight, and stem-to-leaf ratio peaked in the 7-year-old stand (P < 0.05; Table 1). All these parameter values then decreased with increasing stand age, accompanied by degradation of alfalfa and invasion of native grass species in stands aged 10 years and older. The alfalfa grassland was gradually invaded by native plant species such as S. bungeana, Poa annua, Plantago asiatica, and Sonchus oleraceus over time (Fig. 1). Alfalfa AGB initially increased with increasing stand age and reached a peak in the 7-year-old stand (P < 0.05; 679.91 ± 109.50 gm−2), which was significantly higher than that of 10, 15, 20 and 30-year-old stands (P < 0.05; Fig. 2). The growth rate of AGB nonlinearly changed with stand age. Consistent with AGB, the growth rate of AGB was also highest in the 7-year-old stand (P < 0.05; 119.47 ± 48.21 gm−2yr−1), whereas the lowest rate was observed in the 10-year-old stand (P < 0.05; −130.54 ± 33.64 gm−2yr−1). From 15th year, the growth rate of AGB tends to flatten out.

Table 1 Changes in alfalfa productivity across different age groups of reconstructed grassland.

Stand age (year)	Production performance of alfalfa	
	Height (cm)	Stem diameter (cm)	Fresh weight (kg hm−2)	Dry weight (kg hm−2)	Stem-to-leaf ratio (%)	
1	25.50 ± 3.13d	1.97 ± 0.15e	10107.92 ± 322.07bc	2901.28 ± 112.11bc	0.82 ± 0.09b	
5	90.60 ± 7.22b	3.02 ± 0.18ab	8110.12 ± 277.89bc	2910.30 ± 190.50bc	1.06 ± 0.11b	
7	104.00 ± 6.68a	3.62 ± 0.23a	16810.18 ± 586.17a	5340.83 ± 90.80a	2.61 ± 0.24a	
10	30.15 ± 5.57d	2.50 ± 0.31d	1808.22 ± 622.09cd	502.33 ± 37.66cd	0.80 ± 0.08b	
15	76.12 ± 10.56c	3.00 ± 1.00bc	10099.71 ± 278.30bc	4021.05 ± 99.75ab	1.10 ± 0.22b	
20	80.30 ± 12.32c	3.40 ± 1.12a	2507.53 ± 823.16bcd	1100.88 ± 107.82cd	1.62 ± 0.48b	
30	70.45 ± 6.24c	2.98 ± 0.79bc	5514.27 ± 123.74bcd	506.56 ± 70.50d	1.10 ± 0.29b	
Notes.

Different letters in a column indicate significant difference in group means among treatments at the 0.05 level.

Figure 1 Alfalfa growth and vegetation succession along an age gradient of reconstructed grassland.

Figure 2 Alfalfa above-ground biomass (AGB) and growth rate® in different age groups of reconstructed grassland.

Different letters above the error bars indicate significant difference in group means among treatments at the 0.05 level.

Vertical distribution of soil water storage and its feedback to above-ground biomass

Two-way ANOVA showed that stand age, soil depth, and their interaction exhibited a significant effect on SWS distribution (Table 2). The SWS of 0–100 cm soil profile nonlinearly changed with stand age (Fig. 3). The SWS of 0–100 cm soil layers in the 10-year-old stand (312.77 ± 28.56 mm) was the highest, whereas the SWS of 0–100 cm soil layers in the 15-year-old stand (170.78 ± 17.75 mm) was significantly lower than that of other stands. In particular, the lowest SWS occurred in the 80–100 cm soil layer of 15-year-old stand (21.09 ± 8.11 mm; P < 0.05), which was 69.83% lower (P < 0.05) than that of natural grassland (69.9  ± 11.20 mm). The differences in SWS among the various age groups were greatest at the depth range of 80–100 cm where the lowest SWS was recorded in the 15-year-old stand. We collected precipitation data within the 30 years (Fig. S2), which proved that SWS was not effected by precipitation. For alfalfa AGB, it peaked at 7-year-old stand and had the lowest growth rate at 10-year-old stand, and then meet a small peak at 15-year-old stand. Overall, there was a completely negative feedback relationship between the dynamics of AGB and SWS along the 30-year grassland age gradient (Fig. 4A).

Table 2 Two-way ANOVA results for the effects of stand age, soil depth, and their interactions on soil water storage (SWS), organic carbon storage (SOCS), and carbon–water coupling coordination degree (D).

Variable	BD	SWS	SOCS	D	
Factor	F	P	F	P	F	P	F	P	
Stand age	24.46	<0.01	260.62	<0.01	55.12	<0.01	281.47	<0.01	
Soil depth	4.14	<0.05	348.02	<0.01	178.18	<0.01	9.31	<0.01	
Age*Depth	4.60	<0.01	31.59	<0.01	1.91	<0.05	11.03	<0.01	

Figure 3 Distribution of soil water storage (SWS; top) and organic carbon storage (SOCS; bottom) in the 0–100 cm soil profile of grassland stands with different ages.

NG means natural grassland. Different letters near the error bars indicate significant difference in group means among treatment at the 0.05 level.

Figure 4 Relationships between soil water storage (SWS; A), organic carbon storage (SOCS; B), and vegetation above-ground biomass (AGB) in the 0–100 cm soil profile of grassland stands with different ages.

Shaded areas represent two standard errors in each direction.

Vertical distribution of soil organic carbon storage and its feedback to above-ground biomass

Based on the results of two-way ANOVA, there was a significant effect of stand age, soil depth, and interaction between age and depth on SOCS (Table 2). The total SOCS of 0–100 cm soil profile was significantly higher in the 1-, 5-, 7-, 10- and 15-year-old stands than in the 20- and 30-year-old stands (Fig. 3). In particular, the 5-year-old stand had the largest SOCS (P < 0.05; 26.51 ±1.67 kg m2), whereas the 30-year-old stand produced the lowest SOCS (P < 0.05; 14.78 ± 2.41 kg m2). In all cases, the total SOCS of reconstructed grassland was higher than that of natural grassland (18.10 ± 1.33 kg m2). The SOCS exhibited a downward trend toward deeper soil layers in alfalfa grassland stands of different age groups (Fig. 3). For example, the mean SOCS of the 0–20 cm soil layer was 22.2 ± 3.9% higher than that of the 20–40 cm soil layer. A similar difference (19.5 ± 2.4%) was observed in the mean SOCS between 20–40 cm and 40–60 cm soil layers. The effect of soil depth on SOCS diminished in deeper soil layers, as demonstrated by smaller differences between adjacent soil layers (<15%). Within the first 10 years, the AGB of alfalfa and the SOCS of 0–100 cm soil profile showed completely opposite variation, whereas a positive feedback relationship was observed between them from the 15th year (Fig. 4B).

Coordination of carbon–water coupling along a gradient of stand age

Two-way ANOVA revealed a significant effect of stand age, soil depth, and the interaction between age and depth on the D value (Table 2). In the 0–20 cm soil layer, the D values were higher than those of other soil layers in the 5- and 15-year-old stands, and there was a trend of soil carbon–water decoupling from the 18th year (D < 0.6). In the 20–100 cm soil layers, the 15-year-old stand had the lowest D values compared with other age groups. Importantly, the D value of the 80–100 cm soil layer indicated soil carbon–water decoupling from the 12th year (D < 0.6; Fig. 5).

Figure 5 Dynamics of soil carbon–water coupling coordination degree (D) in the 0–100 cm soil profile along an age gradient of reconstructed grassland. The dotted lines indicate the year of soil carbon–water decoupling (D< 0.6).

Discussion

Effect of alfalfa planting on soil water storage

After planting, alfalfa productivity peaked in the 7th year, and the reconstructed grassland was invaded by native grass species from the 10th year. This data would restrict by geography, as alfalfa productivity peaked in the 2nd year and decreased at 4th year in Dongbei (Wang et al., 2022), and decreased at 6th year in Pengyang (Ma et al., 2024). This could contribute to the alfalfa lifespan and soil conditions including soil properties (Li et al., 2021), rhizosphere microbiomes (Ma et al., 2024) and metabolism (Fu et al., 2023). Since vegetation growth is sensitive to soil moisture in arid and semi-arid areas under water-limited conditions, their relationship may be interactive (Wei et al., 2022). In the present study, we found that the AGB and SWS in the 0–100 cm soil profile showed contrasting patterns of temporal variation along an age gradient of reconstructed grassland. The possible reason is that the high productivity and deep root distribution of alfalfa could result in a considerable rise in evapotranspiration, thereby negatively affecting SWS (Li et al., 2021). Additionally, the deep roots of alfalfa could increase water vapor transfer channels by improving soil porosity and aggregate structure (Banwart et al., 2019). This would in turn accelerated soil evaporation, thereby decreasing SWS, especially in deeper soil layers (e.g., 80–100 cm).

The reconstructed alfalfa grassland maintained the highest SWS in the 10th year. At this stage, the degeadation of alfalfa and invasion of native plants was likely to reduce the water demand for plant growth (Wang, Shao & Liu, 2012). As a matter of fact, native plants usually consume less soil water than introduced vegetation species, and the productivity of invaded native plants is relatively low at the early successional stage (Wang, Wu & Zhu, 2014). In contrast, the lowest SWS was recorded in the 15-year-old stand. This might be attributed to the higher AGB of alfalfa, which consumed more soil water and increased soil porosity through continuously developed roots. Another possible reason is that the invaded native plants increasingly developed at this stage, thus consumed a large amount of soil water. Wang et al. (2020) also observed that after alfalfa planting, the SWC in sand-binding areas was lowest after 10–20 years. From the 20th year, the SWS exhibited an upward trend in the reconstructed alfalfa grassland, consistent with the results of Zhang et al. (2018) obtained in northern Loess Plateau. This pattern could be attributed to the decreased water consumption by degraded alfalfa based on lower AGB, as well as the improvement of soil quality and ecosystem stability (Wang, Shao & Liu, 2012).

Despite its increase at 20–30 years, the SWS of reconstructed alfalfa grassland was still lower than that of natural grassland. Continuous planting of alfalfa for six years were reportedly aggravated soil water deficits, which could cause a reduction in crop yield (Huang et al., 2018). Therefore, it is crucial to balance SWS and alfalfa productivity, especially in arid and semi-arid areas like the Loess Plateau. Our results indicate that for alfalfa production and ecological restoration in the semi-arid study area, the optimal age of reconstructed grassland is 7 years from the SWS perspective. Irrigation is needed in the 7th year to prolong the life of alfalfa community and maintain the soil water status. Otherwise, a severe soil water deficit would occur after alfalfa planting for 15 consecutive years. Huang et al. (2018) found that in an arid area of the Loess Plateau, the optimal grassland age was four years, based on the highest alfalfa yield and best soil water conditions. Taken together, the results indicate that planting alfalfa may lead to a severe water deficit in arid areas three years earlier than in semi-arid areas, which warrants further study.

Effect of alfalfa planting on soil organic carbon storage

Irrespective of stand age, the mean SOCS in the 0–100 cm soil profile was higher in the reconstructed grassland than in the natural grassland. This suggests that alfalfa planting has a positive effect on SOC sequestration in the semi-arid study area. According to Xu et al. (2024), nitrogen-fixing legumes such as alfalfa can slow down the decomposition of old and new carbon under vegetation restoration, consequently promoting SOC sequestration. These legumes may also contribute to additional carbon sequestration through an input of biologically fixed nitrogen. The SOCS showed a downward trend with increasing soil depth in the reconstructed grassland. This phenomenon is probably a result of increased surface litter decomposition, which considerably supplemented the SOCS in surface soil (Zhang, Wang & Ai, 2020). Additionally, surface carbon accumulation could occur due to the rhizosphere priming effect, because root penetration stimulates organic carbon mineralization in the deep soil (Shahzad et al., 2018).

Some studies have shown that the initial loss of SOCS in younger plantations is followed by a gradual return in medium-aged plantations (Liu et al., 2017; Laganiere, Angers & Pare, 2010). In the earlier restoration stages (1–10 year), the lower productivity of new vegetation may be responsible for the initial decrease in SOCS (Liu et al., 2017; Zhang et al., 2010; Don et al., 2009). In contrast, we provided strong evidence for negative feedback between SOCS and AGB in the reconstructed alfalfa grassland within the first 10 years (vigorous growth stage of alfalfa). Terrer et al. (2021) also observed a partial trade-off between increased plant growth and SOCS, whereby ecosystems with plant growth under greater nutrient limitation accumulate more carbon below ground. This trade-off may be associated with plant nutrient acquisition, as alfalfa produces biomass by adsorbing soil nutrients and thereby decreases SOCS.

Interestingly, the feedback relationship between SOCS and AGB turned out to be positive after alfalfa planting for more than 10 years. SOCS reached a peak in the 10th year, which could be partly attributed to the succession of reconstructed grassland. The invasion of native grass species sharply decreased plant nutrient acquisition and consequently eliminated the trade-off between SWS and SOCS. Furthermore, a higher level of plant diversity at this successional stage could increase SOCS (Chen et al., 2018; Prommer et al., 2019). From the 10th year onwards, the SOCS decreased over time, most likely a result of grassland succession coupled with the allelopathic effect of alfalfa. Alfalfa can release allelochemicals into the soil through root exudates, which accumulate over time (Li, Liang & Jiang, 2004). Some root exudates liberate organic compounds from protective associations with minerals, leading to SOCS loss (Keiluweit et al., 2015). Considering the trade-off between AGB and SOCS, carbon fertilization may be necessary in the 7th year of alfalfa planting, based on the largest crop yield and lowest SOCS within the first 10 years.

Coupling between soil water and organic carbon storage

The coupling between SWS and SOCS drives ecosystem stability and sustainability in arid and semi-arid areas (Li et al., 2021). When SWS and SOCS promote each other, the ecosystem maintains orderly and positive development with enhanced functions (benign coupling: D ≥ 0.6). When the two soil factors interfere with and disrupt each other, the ecosystem develops in a negative direction with attenuated functions (vicious coupling: D < 0.6) (Liang et al., 2022; Li et al., 2021). Results of the present study showed that there was a benign coupling between the mean SWS and SOCS of 0–100 cm soil profile within the first 10 years of alfalfa planting. During this stage, both SWS and SOCS occurred at higher levels than those of other successional stages. Soil water and SOC could interact with each other through multiple mechanisms. On the one hand, SWC and soil water movement affect soil processes, including microbial activity, mineral leaching, and biogeochemical cycles (Manzoni & Porporato, 2009). These processes determine the availability of soil nutrients (Otieno, K’otuto & Maina, 2010) and ultimately influence vegetation productivity and organic matter return (Li et al., 2021). For example, Linkosalo, Kolari & Pumpanen (2013) found that high SWC improved soil microbial growth and activity by enhancing substrate and oxygen diffusion, which in turn increased organic carbon decomposition. On the other hand, SOC plays a key role in modifying soil physical properties, improving soil chemical quality, and preserving plant nutrients (An et al., 2010). Soil structural improvement due to increased SOC also influences soil pore geometry and size distribution, which then controls soil water-holding capacity (Wang et al., 2015).

After 12–18 years of alfalfa planting, SWS and SOCS showed a vicious coupling in the reconstructed grassland. At this stage, soil carbon–water coupling might be interfered by low SWS due to high water consumption by vegetation in the water-limited study area (Gao, Meng & Zhao, 2017). The unstable state of soil carbon–water coupling in the arid and semi-arid areas of Loess Plateau was caused by the substantial invasion of native pioneer plants and the accelerated rate of vegetation community succession (Liang et al., 2022; Li et al., 2021). In the 20th year, SWS and SOCS showed a decoupling, which ultimately changed to a vicious coupling in the 30th year. This shift could be attributed to the recovery of SWS, but further studies are still needed to verify the specific mechanisms. The coordination degree of soil carbon–water coupling was less disturbed in the surface soil (0–20 cm) than in the subsurface soil (20–100 cm). This pattern could be related to the surface aggregation of SOC (Zhang, Wang & Ai, 2020) and less water loss in shallow soil. Based on the results, alfalfa should be planted for no more than 12 consecutive years in the study area to maintain a balanced soil carbon–water coupling and ensure sustainable ecological service provision.

Conclusions

Long-term planting of alfalfa led to the loss of SWS and the increase of SOCS in a semi-arid area of the Loess Plateau. Alfalfa productivity showed a negative feedback relationship with SWS over a 30-years reconstructed chronosequence, whereas a similar relationship with SOCS was observed only in the first 10 years. SWS and SOCS showed a benign coupling between 1–10 years and were decoupled from the 12th year, with a vicious coupling in the 15th year. With respect to agricultural production, it is recommended to supply a greater amount of water and organic fertilizer from the 7th year of alfalfa planting, and reseeding may be necessary around the 10th year to prolong the life of alfalfa community. From the ecological function perspective, alfalfa should be planted for no more than 12 years to maintain the soil water status, organic carbon sequestration capacity, and benign carbon–water coupling.

Supplemental Information

Figure S1 Study area in the loess hilly region of Northwest China (Longde County, Ningxia)

Image source: http://bzdt.ch.mnr.gov.cn/index.html.

Figure S2 Annual precipitation data of the stand age (year)

Figure S3 Distribution of soil water storage (SWS) in the 0–100 cm soil profile of grassland stands with different ages

NG means natural grassland. * indicate significant difference in group means among treatment at the 0.05 level.

Table S1 Major vegetation species and their importance values in reconstructed grassland stands with different ages

Data S1 Raw data of soil BD, SW and SOC

Additional Information and Declarations

Competing Interests

Author Contributions

Data Availability

The authors declare there are no competing interests.

Yuanyuan Ma analyzed the data, prepared figures and/or tables, authored or reviewed drafts of the article, and approved the final draft.

Xiaoping Zhou analyzed the data, prepared figures and/or tables, authored or reviewed drafts of the article, and approved the final draft.

Yan Shen conceived and designed the experiments, performed the experiments, authored or reviewed drafts of the article, and approved the final draft.

Hongbin Ma conceived and designed the experiments, performed the experiments, authored or reviewed drafts of the article, and approved the final draft.

Bingzhe Fu performed the experiments, authored or reviewed drafts of the article, and approved the final draft.

Jian Lan performed the experiments, authored or reviewed drafts of the article, and approved the final draft.

The following information was supplied regarding data availability:

The raw measurements are available in the Supplementary Files.

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
