# Peer review of "Long-term alfalfa planting mediates the coupling of soil water and organic carbon storage in a semi-arid area of the Loess Plateau, China"

_PeerJ, doi:10.7717/peerj.18373_

## Round 0.1 · original submission · Major Revisions

· Academic Editor

Major Revisions

I ask you to carefully correct the shortcomings of the manuscript. Some results raise questions about their reliability. For example, the partial lack of statistical processing in Figure 2 or the same values ​​of the standard deviation for all columns of Figure 2. I hope that careful analysis of each of the reviewers' comments will help you to succeed and publish this manuscript.

Reviewer 1 ·

Basic reporting

The language of the manuscript is clear and sufficiently accurate, with good references to relevant literature in the field in the preface and discussion sections of the manuscript, and the structure of the manuscript is fine, with some of the images and tables needing to be adjusted to fit. The conclusion of the manuscript also responds well to the scientific questions in the preface of the manuscript.

Experimental design

The theme of the manuscript is aligned with the objectives and framework of the journal and has a good fit. The experimental design of the manuscript needs further refinement and clarification, thus strengthening the scientific validity and credibility of the results and conclusions of the manuscript; the research scientific questions are clear and important. The experiment is well-documented and informative and can provide good support for the reliability of the manuscript's conclusions.

Validity of the findings

This manuscript analyzes the water-carbon balance of alfalfa over a 30-year growing period with the help of the “space-for-time” method. The authors analyze the temporal dynamics of key indicators such as stratified soil water content (SWS), above-ground biomass (AGB), soil organic carbon content (SOCS), etc., and find the optimal year of fertilizing, the optimal year of replanting, and the maximum yielding planting year for alfalfa. The key values such as the optimal fertilizer year, the optimal replanting year, and the maximum yield planting year of alfalfa were found. The results have good reference significance and scientific value for the restoration of saline-alkaline land and pasture planting in arid/semi-arid areas. Therefore, I recommend that the manuscript continue to undergo minor revisions to meet the open publication standard.

Additional comments

The following are my detailed revisions for the content of the manuscript:

1. Is it a coincidence that all of the years in which your alfalfa yields shifted were years in which you sampled? Is this a coincidence, or is there a problem with your current analytical methodology or the way you fit the trend to the sampled data?
2. Do the data from the sampling time points in the manuscript truly reflect the real growth process of alfalfa? Are samples from different growth years more influenced by the sample site? What were your criteria for the selection of sampling time points?
3. In lines 114-116, authors are requested to give a map of the spatial layout of the sampling sites, preferably with topographic data or the latest imaging data of the study area as the map, in order to improve the readers' understanding of the situation of the sampling sites in the study, and also as a prerequisite for the data of the study to have a cross-comparison. The map can form a combined map with Figure 1 in the current manuscript or stand alone. Also, the authors are requested to give the relevant citations of the sample site as a long-term monitoring and research to illustrate the authenticity and scientific validity of the sample site of the study.
4. in line 165, α=β= 0.5 in this study,where does that weight come from? What is the basis and is it consistent with the reality of alfalfa growth mechanism?
5. 183 Why is there a trough in alfalfa above-ground biomass in year 10 in line 2? Is it an environmental (soil or climate) effect of the sampling area? Or is it a data problem, and if it is a data problem, does it affect the year in which the maximum aboveground biomass occurs? For example, might the maximum occur in year 10 instead of year 7?
6. The labeled text in Figure 4 is too small to be legible, please adjust it to the right size to ensure that every text on the figure is legible. And adjust the labeling position of the sub-figures to make them uniform.
7. In Figure 4A, it is recommended to add the annual precipitation data of the sampling year to exclude that the change of soil water content is not from the change of precipitation. This is because in arid zones, changes in soil water content are most likely to come from the effects of changes in precipitation, not simply changes in water use by the vegetation root system.
8. In line 217, note the way the layers are written for the corresponding year.

·

Basic reporting

No comment.
Pls see more detial in our comments.

Experimental design

No comment.

Validity of the findings

No comment.
Pls see more detial in our comments.

Additional comments

Comments:
Abstract:
Line 31: pls double check the P value of 0.2569.

Introduction
Line 40: It is worth noting that arid and semi-arid areas are very different in farming system management. Please explain why you only focused on one are. Over last 10 years, we also did some work on the Loess Plateau. I would recommend the authors to go through the following publications: “Climate warming suppresses abundant soil fungal taxa and reduces soil carbon efflux in a semi‐arid grassland”, “Moderate precipitation reduction enhances nitrogen cycling and soil nitrous oxide emissions in a semi‐arid grassland”, and Mowing alters nitrogen effects on the community-level plant stoichiometry through shifting plant functional groups in a semi-arid grassland although we focused more on climate change and microbes.
Line 48: It should be resistant rather than tolerant. For alfalfa drought-resistance is the more important trait than drought tolerance.
Line 63-65: For the sentence of "Owing to increasing water demand, continuous planting of alfalfa causes declines in shallow groundwater levels", did the authors mean “Does water consumption continue to increase? After planting of more than a few years, water consumption decreases due to the gradual alfalfa death?” I am a bit confused.
Line 71-75: Did the authors mean “planting alfalfa restores degraded grassland to improve SOCS?” Please use these words "revegetation" or "reconstructed grasslands" consistent in the whole manuscript. This part is not very clear.
Line 79-89: The whole paragraph is about the relationship between SWS and SOCS in grassland ecosystems, yet, the authors did not mention alfalfa fields. Pls added more detail.
Line 92-97: "across different age groups of reconstructed grassland" "under revegetation with alfalfa" "age of reconstructed grassland". Suggested to modify the keywords uniformly.
Line 96: what is crop yield? Is it alfalfa forage yield or crop yield after alfalfa?

Materials and methods
Line 109-111: pls add a ref.
Line 124-125: Please explain "given age group" and "younger age group" in detail.

Results
The relationship between SWS and stand age was not analyzed in Fig. 3.
Line 179: did the authors have this data? I am wondering whether the authors had measured the abundance of native grass species, it would be a strong supplement. I also understand the abundance and composition of native grass species may be anther good story for another paper.
Line 202: "in the 1-15-year-old stands"?

Discussion
Line 225-228, These sentences contain too many results and please added more discussion on these results. I would also suggest to delete “these results are compatible with… Li et al. 2005”. You actually did not need to cite a ref published around 20 years ago to support your results.
Line 246-248: “the SWC in sand-binding areas was lowest at the middle successional stage of grassland.” What is the middle successional stage? There is no division of succession stages in the results?
Line 250-251: "This pattern could be attributed to the decreased water consumption by alfalfa based on lower AGB, as well as the improvement of soil quality and ecosystem stability". "Continuous planting of alfalfa reportedly aggravated soil water deficits, which could cause a reduction in crop yield." (L254-255). These two sentences are contradictory or I misunderstood them.
Line 269-270: it is not a good choice to cite these two refs and I would suggest the authors to cite some refs published in high impact journals or in peerj.
Line 274: “supergene” mean?
Line 274: do you mean “increased or increased surface litter decomposition”. It was not clear.
Line 322: here “vicious” mean?

·

Basic reporting

The placement and formatting of formulas and formula serial numbers in the text need to be kept uniform, and modifications are recommended.
The formatting of the paragraphs in the text should be revised in order to maintain uniformity.
Are photos of alfalfa growth and vegetation in year 5 missing from Figure 1?
For figures and tables, please double-check the changes, and the order in which the figures are laid out is suggested to be modified to be more in line with reading habits (e.g., NG, 1yr, 5yr........ and 30yr).

Experimental design

Please add the species of alfalfa to the text.
line 119,Please add detailed data sources, such as: sampling method and sampling quantity.
It is suggested that the P in Table 2 should be uniformly expressed as scope or specific data.
The significance of this paragraph is not reflected in the figure, and it is recommended to supplement the data analysis to support the textual expression (line 190 – 191).
The description of the comparison of each soil layer in the results is too general and would like to be explained in detail(line 184 -185)
Is the 0-5 year feedback relationship reversed (line 207)?
In line 221-224.Is this conclusion restricted by geography? Hope to add a more reasonable explanation for this phenomenon.

Validity of the findings

Lines 214-216, the 0-20 cm soil layer does not have a greater D value in year 10 than the other layers, please check for modifications.
line 106 and 114, the average annual precipitation and distance should be checked.
Inaccurate use of text, professional editing is recommended (e.g. line 17, 23, 36, 71, 72, 76, 96...)
In line 182, biomass is mentioned in the title, but there is no specific analysis of biomass in the paragraph, only relevance is mentioned in the last sentence, and there is no link to the previous outcome analysis. Suggested modifications.
It is recommended to add the P-value after the description of the variance analysis in the outcome analysis.

Additional comments

What was the basis for selecting reconstructed alfalfa grasslands of different stand ages (1, 5, 7, 10, 15, 20, and 30 years) for the study?
Line 187-189, the expression is ambiguous and inaccurate, and changes are suggested. The same issues in the text should be double-checked and corrected.
In line 118,Please explain why the author chose the period?
It is recommended to replace the words that are too redundant in the article and change them to simple words (e.g., line 26, 74, 92)

---

## Round 0.2 · accepted · Accept

· Academic Editor

Accept

Now the article meets modern requirements, and I think that it can be published in the next issue of the journal. The article should be interesting to many specialists from different countries. I hope that you will continue research in this direction.

Reviewer 1 ·

Basic reporting

The revised manuscript is good and reasonably explains the various doubts and concerns that readers may have had in the previous review. The revised manuscript is logically clear and smoothly expressed, and the manuscript images and tables support the presentation and argumentation process of the manuscript well.

Experimental design

The experimental design of the manuscript is reasonable, and the sampling data can illustrate the process of water-carbon changes in alfalfa growth, which is of good value for research in this field.

Validity of the findings

The manuscript provides a good explanation of the water-carbon interactions in the alfalfa growth cycle, which provides useful theoretical support for livestock operations and the achievement of peak carbon goals in the agro-pastoral industry.

Additional comments

No comment.

·

Basic reporting

No comment

Experimental design

No comment

Validity of the findings

No comment

Additional comments

I am happy with the changes done by the authors and enjoy reading the revision. Thanks for the efforts.
I only have some minor points below:

1) As I mentioned in my previouse comments, I would recommend the authors to cite the following publications:“Climate warming suppresses abundant soil fungal taxa and reduces soil carbon efflux in a semi‐arid grassland”, “Moderate precipitation reduction enhances nitrogen cycling and soil nitrous oxide emissions in a semi‐arid grassland” in the first introduction paragraph.

2) Line 72: delete storage

3) contribute not contributed